# ADAM10-Mediated Cleavage of ICAM-1 Is Involved in Neutrophil Transendothelial Migration

**DOI:** 10.3390/cells10020232

**Published:** 2021-01-25

**Authors:** Sofia K. H. Morsing, Timo Rademakers, Sanne L. N. Brouns, Anne-Marieke D. van Stalborch, Marjo M. P. C. Donners, Jaap D. van Buul

**Affiliations:** 1Molecular Cell Biology Lab, Department Molecular and Cellular Homeostasis, Sanquin Research and Landsteiner Laboratory, University of Amsterdam, Plesmanlaan 125, 1066 CX Amsterdam, The Netherlands; s.morsing@sanquin.nl (S.K.H.M.); t.rademakers@maastrichtuniversity.nl (T.R.); sanne.brouns@maastrichtuniversity.nl (S.L.N.B.); ad.stalborch@sanquin.nl (A.-M.D.v.S.); 2Department of Pathology, Cardiovascular Research Institute Maastricht, Maastricht University, P. Debyelaan 25, 6229 HX Maastricht, The Netherlands; 3Leeuwenhoek Centre for Advanced Microscopy (LCAM), Section Molecular Cytology at Swammerdam Institute for Life Sciences (SILS), University of Amsterdam, 1066 CX Amsterdam, The Netherlands

**Keywords:** transmigration, ICAM-1, endothelium, ADAM, shedding

## Abstract

To efficiently cross the endothelial barrier during inflammation, neutrophils first firmly adhere to the endothelial surface using the endothelial adhesion molecule ICAM-1. Upon actual transmigration, the release from ICAM-1 is required. While Integrin LFA1/Mac1 de-activation is one described mechanism that leads to this, direct cleavage of ICAM-1 from the endothelium represents a second option. We found that a disintegrin and metalloprotease 10 (ADAM10) cleaves the extracellular domain of ICAM-1 from the endothelial surface. Silencing or inhibiting endothelial ADAM10 impaired the efficiency of neutrophils to cross the endothelium, suggesting that neutrophils use endothelial ADAM10 to dissociate from ICAM-1. Indeed, when measuring transmigration kinetics, neutrophils took almost twice as much time to finish the diapedesis step when ADAM10 was silenced. Importantly, we found increased levels of ICAM-1 on the transmigrating neutrophils when crossing an endothelial monolayer where such increased levels were not detected when neutrophils crossed bare filters. Using ICAM-1-GFP-expressing endothelial cells, we show that ICAM-1 presence on the neutrophils can also occur by membrane transfer from the endothelium to the neutrophil. Based on these findings, we conclude that endothelial ADAM10 contributes in part to neutrophil transendothelial migration by cleaving ICAM-1, thereby supporting the release of neutrophils from the endothelium during the final diapedesis step.

## 1. Introduction

During inflammation, circulating leukocytes need to cross the vascular endothelial barrier, a process known as transendothelial migration (TEM) [1,2,3]. The three-step paradigm of TEM describes how leukocytes roll, adhere, and migrate through the endothelial barrier, which has been extensively studied [4,5]. However, how leukocytes are released from the endothelium to migrate into the underlying tissue has largely been neglected.

When coming into contact with inflamed endothelium, neutrophils utilize the β2-integrins LFA-1 (CD11a/CD18) and Mac-1 (CD11b/CD18) to firmly bind the endothelial intracellular adhesion molecule ICAM-1 [6]. Whereas Mac-1 supports locomotion on the ICAM-1 surface, LFA-1 specifically seems to mediate firm adhesion [7]. Having penetrated the endothelium, ICAM-1 has been shown to adjoin the transmigrating neutrophil from the apical to the basolateral side [8]. It has also been reported that LFA-1 redistributes to the neutrophil uropod during TEM, indicating that there is still an ICAM-1-LFA-1 interaction in the final steps of diapedesis [9]. For leukocytes to continue their journey into the underlying tissue, they need to be released from the endothelium, and break the ICAM-1-LFA-1 interaction. One possible mechanism for this is integrin deactivation by the neutrophil [10]. The other option is physical shedding of adhesion molecules from the endothelial membrane surface.

A disintegrin and metalloproteinases (ADAMs) are proteases that are involved in the shedding of a large number of cell surface proteins [11]. We and others have previously shown that ADAM10 and 17 can cleave the junctional adhesion molecule vascular endothelial (VE)-cadherin [12,13], which maintains endothelial barrier function and plays a crucial role during paracellular TEM of neutrophils [14]. In vivo findings in mice with a genetic deficiency of ADAM10 in leukocytes showed impaired TEM of leukocytes [15], in line with an in vitro study using small interfering RNA against ADAM10 in T-cells, as well as in endothelial cells by mediating the cleavage of VE-cadherin [13]. ICAM-1 and VCAM-1 are prone to be cleaved from the surface, e.g., by ADAM17 [16,17,18]. However, the functional consequence of ADAM-mediated cleavage of ICAM-1 from the endothelium for leukocyte TEM remains ambiguous.

Here, we found that silencing or inhibiting endothelial ADAM10 increased the surface expression of ICAM-1 and promoted the number of adherent neutrophils. However, despite increased number of adherent neutrophils, silencing of endothelial ADAM10 impaired the efficiency of neutrophils to cross the endothelium. Our data further supported a role for endothelial ADAM10 to dissociate neutrophils from the endothelium upon transmigration. Using Image Stream analysis and the use of an endothelial expressing ICAM-1-GFP construct, we found that endothelial ICAM-1 can also be transferred to the neutrophil through membrane transfer. Based on these findings, we conclude that endothelial ADAM10 contributes to neutrophil TEM by cleaving ICAM-1.

## 2. Materials and Methods

Antibodies, plasmid, and reagents: The following antibodies (Ab) against human origin were used: extracellular targeting rabbit polyclonal Ab (pAb) against ICAM-1 (clone H-108), intracellular goat anti-VE-cadherin (clone C-19), and monoclonal Ab mouse anti-ICAM-1*405 (clone 11C81) were purchased from Santa-Cruz Biotechnology Inc., Dalllas, TX, USA; pAb rabbit against extracellular VE-cadherin (clone JC63.1) was obtained from Cayman, Ann Arbor, MI, USA; mAb mouse against actin (clone AC-40) was purchased at Sigma, St. Louis, MO, USA; mAb mouse against VE-cadherin 647 (clone 55-7H1) was obtained from BD Biosciences, Stockholm, Sweden; pAb rabbit against ADAM10 was purchased at EMD Millipore, Burlington, MA, USA. For actin staining, Phalloidin Texs Red obtained from Invitrogen, Carlsbad, CA, USA, was used. Hoechst 33342 (clone H-1399) was purchased from Molecular probes, Eugene, OR, USA. Secondary HRP-conjugated goat anti-mouse, swine anti-rabbit, and rabbit anti-goat were purchased from Dako, Ely, United Kingdom. The metalloproteinase inhibitor GI254023X was a kind gift from GaxoSmithKline, Brentford, United Kingdom. The ICAM-1-GFP construct, where the GFP tag was fused to the C-terminus of ICAM-1, meaning that GFP is present at the intracellular tail of ICAM-1, was a kind gift from Dr. Francisco Sanchez-Madrid (Madrid, Spain) [19,20].

Short hairpin constructs: To knockdown ADAM10 in HUVECs, we used short hairpin RNAs (shRNAs) in pLKO.1 (TRC cloning vector) targeting ADAM10 (TRCN6675; Sigma) or non-targeting shRNA (CTRL, SHC002). Plasmid DNA purification using NucleoBond Maxi was performed according to manufacturer’s instructions (Macherey-Nagel, Düren, Germany) and DNA concentration was kept between 900 and 1200 ng/μL. To produce high titer lentivirus stocks, HEK293T cells were transfected with ADAM10 or CTRL DNA in Dulbecco’s Modified Eagle’s Medium (IMDM, Lonza, Basel, Switzerland) supplemented with 10% fetal calf serum (FCS), 2 mM L-Glutamine, 100 U/mL Penicillin, 0.1 mg/mL Streptomycin and 1 mM Sodium Pyruvate. Transfection was performed by addition of a viral construct mastermix (44 ng/cm^2^ pHDMG-G VSV ENV, 22 ng/cm^2^ pHDM-HgpM2 GAG/POL, 22 ng/cm^2^ pRC-CMV-Rev1b REV, 22 ng/cm^2^ pHDM-TAT 1B) to TransIT-LT1 transfection reagent (0.4μL/cm^2^, Mirus Bio, Madison, WI, USA) which was diluted in Optimem (10 μL/cm^2^, Life Technologies, Carlsbad, CA, USA). This mix was then combined with the DNA constructs (289 ng/cm^2^) and added to the HEK293T cells. IDMEM containing produced lentivirus was harvested twice after 24 h and 48 h, pooled, filtered over 0.45 μM pore filter, and stored at −80 °C until use.

Cell culture: HUVECs (Lonza, Basel, Switzerland) were cultured in endothelial growth medium supplemented with single quots (EGM2, Lonza) and seeded on fibronectin-coated culture dishes. Human embryonic kidney 293T cells (HEK293T) were cultured in IMDM^++^ and seeded on fibronectin and gelatin co-coated culture dishes. All cell lines were cultured at 37 °C and 5% CO_2_. For passaging, cells were washed in Phosphate-buffer Saline (PBS) and dissociated from the plate with Trypsin/EDTA (Sigma). To neutralize Trypsin/EDTA, trypsin neutralizing solution (TNS; Lonza, Basel, Switzerland) was added in a 1:1 Ratio to Trypsin/EDTA dilution.

Where indicated, inhibitor for ADAM10 (GI254023X, indicated as GI), was diluted in EGM2 or serum free EGM2 and incubated for 30 min or 24 h at a concentration of 5 μM. Transduction with lentivirus containing short hairpin constructs of interest was for at least 48 h. An inflammatory phenotype was induced by stimulating cells for the given time points with TNFα (10 ng/mL, PeproTech, Rocky Hill, NJ, USA) together with inhibitors (where applicable).

Western blotting: To assess cleavage of adhesion molecules and to confirm efficient knockdown, Western blots were performed. Cells were grown until 70–80% confluence and TNFα was added in serum free medium 24 h before lysing using SDS sample buffer (Invitrogen). Samples were loaded on 10% polyacrylamide gels (PAGE; 5 mL Tris A, 6.6 mL Acryl Bis (National Diagnostics, Brunstatt-Didenheim, France), 8.6 mL H_2_O, 20 μL TEMED (Invitrogen, Carlsbad, CA, USA), and 100 μL 10% APS) and were separated at 120 V. Proteins were transferred to a nitrocellulose membrane (GE Healthcare, Chicago, IL, USA) by blotting for 2 h at 300 mA. After transfer, membranes were blocked in 5% bovine serum albumin (BSA, Affimetrix, Santa Clara, CA, USA) or 5% milk powder for 1 h at room temperature. Staining of blots was performed with primary antibodies against extracellular ICAM-1 (1:1000), and VE-cadherin (1:2000) overnight at 4 °C. Membranes were washed 3 times in TBST and incubated with secondary antibodies labelled with horse radish peroxidase (HRP, anti-rabbit and anti-goat, all 1:5000 diluted). Visualization of proteins was performed using (super) enhanced chemiluminescence (ECL, Pierce, Waltham, MA, USA). To control for equal loading, blots were stained for actin (1:5000) and secondary anti-mouse-HRP (1:5000). Intensities of bands were quantified using ImageJ software.

Neutrophil isolation: Human polymorphonuclear cells (PMN) were isolated according to Roos and De Boer [21]. In brief, from heparin anti-coagulated whole blood from healthy donors using a Percoll gradient (1.076 g/mL). Tubes were centrifuged (Rotanta 96R) at 2000× *g* rpm with a slow start and low brake for 20 min. Erythrocytes were lysed in a buffer containing 4.15 g NH_4_Cl, 0.5 g KHCO_3_, and 18.5 mg EDTA (triplex III) diluted in 500 mL demi water and centrifuged at 1500× *g* rpm for 5 min at 4 °C. After lyses, neutrophils were washed in ice-cold PBS, centrifuged again at 1500× *g* rpm for 5 min at 4 °C, and resuspended in HEPES^++^ (1 mM CaCl_2_, 0.5% human serum albumin from a 200 g/L stock, and 0.1 g glucose in 100 mL HEPES buffer containing 132 mM NaCl, 20 mM HEPES, 6 mM KCl, 1 mM MgSO_4_, 1.2 mM K_2_HPO_4_·3H_2_O, pH 7.4) and PMN concentration was determined using a Casy counter (Roche Innovatis). Neutrophils were diluted in HEPES^++^ in a concentration of 5 × 10^6^ neutrophils/mL [22] Isolated neutrophils were kept at room temperature until use. This neutrophil isolation method typically yields >95% purity.

Neutrophil transendothelial migration (TEM): Transendothelial migration experiments under flow conditions were performed according to Kroon et al. [23]. In brief, TEM under flow was performed to mimic physiological conditions of the human blood vessel. 100,000 HUVECs were seeded and cultured in FN-coated Ibidi μ-slide VI0.4 (Ibidi) until confluency, after which they were stimulated overnight with TNFα (10 ng/mL) to mimic inflammatory conditions. Where indicated, inhibitors were added 30 min before the addition of neutrophils to examine transendothelial migration under flow. 200 μL of neutrophils in HEPES^++^ were incubated at 37 °C for 30 min. Ibidi flow chambers were connected to a perfusion system and HEPES^++^ was flowed at a rate of 1 dyne/cm^2^. Neutrophils were injected into the perfusion system via an injector port and real-time imaging was performed with a widefield Zeiss Axiovert microscope (Zeiss) with 10× objective. Experiments were performed at 37 °C and 5% CO_2_ [22]. After 20 min of neutrophil perfusion, the experiment was stopped, and flow chambers were fixed for 5 min in 3.7% formaldehyde (Sigma) diluted in PBS containing 1 mM CaCl_2_ and 0.5 mM MgCl_2_ for immunofluorescence to stain for proper monolayer integrity. Numbers of adhering and transmigrated neutrophils were counted manually using ImageJ Cell counter plug-in with DIC images and percentages of transmigrated neutrophils were calculated.

Flow cytometry: To assess surface expression of adhesion molecules, we performed FACS analysis. HUVECs were rinsed in PBS and detached using Accutase (GE Healthcare). Detached cells were resuspended in ice-cold EGM2 and centrifuged at 1200 rpm for 5 min. Cells were incubated with fluorescently labelled antibodies directed against ICAM-1 (AF405-labeled, 1:100), and VE-cadherin (AF647-labeled, 1:100). Antibodies were diluted in PBS + 0.5% BSA (PBS/BSA) and incubated for 30 min on ice. After labeling, cells were washed twice in PBS/BSA and centrifuged at 1200 rpm for 5 min in between wash steps. Fluorescent measuring was performed using a 3 L Canto FACS (BD Biosciences). HUVECs were gated on their size (FSC-A vs. SSC-A). Analysis of mean fluorescent intensities were performed with BD FACS Diva (BD Biosciences).

Neutrophil migration speed, distance, and velocity: The number of adherent neutrophils was evaluated using the “MTrackJ” command in ImageJ software (National Institutes of Health, Bethesda, Maryland, MD, USA). This software including assistance option is freely available using this link: https://imagescience.org/meijering/software/mtrackj/.

ELISA—To confirm cleavage of adhesion molecules, ELISA kits for soluble ICAM-1 was used (Diaclone). Supernatants of cell culture medium were centrifuged for 10 min at 1000 G and diluted 1:100/1:50 respectively in standard diluent. Wells pre-coated with mAbs against ICAM-1 was incubated with standard (known concentration), blank, or supernatant samples and labelled with biotinylated-anti-ICAM-1 for 1 h. After 3 times washing, streptavidin-horseradish peroxidase (HRP) was added and incubated for 30 min. Samples were washed again and tetramethylbenzidine (TMB), a substrate for HRP, was incubated for 15–20 min. H_2_SO_4_ was added to stop the reaction and absorbance was measured at 450 nm using a microplate reader (Infinite M200, Tecan, Männedorf, Switzerland). Standard curves were generated, and protein concentrations were calculated.

Transwell assay: To confirm whether knockdown of ADAM10 alters cleavage of adhesion/junction molecules, resulting in less transferring of these molecules towards neutrophils during transendothelial migration, Transwell systems were used. Transwell filters with a 5.0 μm pore size (Costar) were coated with fibronectin. 100,000 HUVECs were seeded on Transwell filters, grown until confluency, and stimulated overnight with TNFα (10 ng/mL). C5A (10 nM, Sigma), a chemoattractant, was diluted in HEPES^++^ and added to the bottom compartment, and 5 × 10^5^ of neutrophils were added on top of the filter. After 1 h, HEPES^++^ containing neutrophils from the bottom compartment was centrifuged at 1200× *g* rpm for 5 min to collect cell pellets. Neutrophils were incubated with fluorescently labelled antibodies directed against ICAM-1 (AF405/FITC-labeled, 1:100), and VE-cadherin (AF647-labeled, 1:100). Staining of neutrophils was performed by incubating antibodies diluted in PBS/BSA for 30 min on ice. After labeling, cells were washed twice in PBS/BSA and centrifuged at 1200× *g* rpm for 5 min in between. Transwell filters were fixed in 3.7% formaldehyde and stained for Phalloidin and Hoechst to check monolayers for confluency. Measurements of neutrophils were performed using a 3 L Canto FACS (BD Biosciences), 5 L FACS Fortessa (BD Biosciences) and data were analyzed using BD FACS Diva (BD Biosciences).

Image Stream analysis: Analysis of imaging flow cytometry data was evaluated by an imaging flow cytometer (Image Stream, Amnis, Seattle, WA, USA) acquiring 5000 images in the bright field channel (DIC) and the fluorescence emission channels at 660–740 nm (ICAM-1 antibody in red) and 480–560 nm (GFP/YFP signal) for each sample. The acquired images were analyzed with IDEAS software (Amnis).

Statistics: Statistical analysis was performed using Graphpad Prism 8.0 software (Graphpad Software). Significant differences compared to control conditions were performed by non-parametric Kruskal–Wallis with Dunn’s multiple comparison test or Mann–Whitney U tests, since we assumed our small sample size (*n* = 3–5) to be not normally distributed. Paired *t* tests were used for statistical analysis of normalized fold changes of short hairpin data. *p*-values < 0.05 were statistically different.

## 3. Results

### 3.1. Inhibition of ADAM10 Reduces ICAM-1 Cleavage and Impairs Neutrophil TEM

ADAM10 plays a crucial role in leukocyte recruitment into inflamed tissue. Although it is known to cleave endothelial junctional molecules, its role in cleavage of endothelial adhesion molecules is unclear. Therefore, we pretreated HUVECs with the ADAM10 inhibitor GI254023X [24]. As previously shown by us and others [12,13], ADAM10 inhibition almost completely blunted VE-cadherin cleavage, though full length VE-cadherin in the cell lysate remained unaffected (Figure 1A,B). Interestingly, in TNFα-treated HUVECs, to mimic inflammation, inhibition of the cleavage activity of ADAM10 reduced the levels of soluble ICAM-1 in the cell culture medium. Though not statistically significant when assessing with Western blotting (Figure 1C,D, *p* = 0.097), quantitative protein determination using ELISA (Figure 1E) indicated that ADAM10 inhibition significantly decreased cleavage of endothelial ICAM-1 under inflammatory conditions.

To investigate the functional contribution of endothelial ADAM10 in neutrophil TEM, migration experiments under flow conditions were performed. Confluent TNFα-treated HUVECs were pre-incubated for 30 min with the ADAM10 inhibitor and subsequently perfused with neutrophils. Inhibition of ADAM10 showed a significant enhanced neutrophil adhesion. However, also neutrophil transendothelial migration (TEM) at 5 and 10 min was reduced, albeit not to a significant level. These differences disappeared at 20 min of perfusion (Figure 1F–H). Adherent neutrophils appeared as bright dots on the endothelium, whereas transmigrated neutrophils displayed a dark grey phenotype (Appendix A). We concluded that the reversible nature of the ADAM10 inhibitor, which was not present during leukocyte perfusion, may have been responsible for the limited effect. However, the GI inhibitor has a half maximal inhibitory concentration (IC) for ADAM10 of 5.3 nM but can also inhibit ADAM17 with an IC50 of 541 nM [25]. Thus, the concentration used here may also affect ADAM17 activity.

To overcome this issue, we silenced ADAM10 in the endothelial cells using shRNA (Figure 2A) and confirmed that also under these conditions, reduced levels of soluble ICAM-1 and VE-cadherin were detected. Shedding of VE-cadherin was used as a positive control and showed that silencing of ADAM10 reduced the levels of soluble VE-cadherin in the supernatant (Figure 2B). Under the same conditions, reduced levels of soluble ICAM-1 were detected using Western blotting, albeit not statistically significant (Figure 2C). Alternatively, we used ELISA to measure soluble ICAM-1 under ADAM10 knockdown and control conditions. ADAM10 knockdown showed a significant decrease in the amount of soluble ICAM-1 in the supernatant (Figure 2D).

In line with ADAM10 inhibition, knockdown of ADAM10 resulted in an increase in neutrophil adhesion under flow, although only the 10-min time point showed a significant increase (Figure 2E,F). We found a significant reduction of neutrophil TEM (Figure 2E,G). However, as with the inhibitor studies, at 20 min of perfusion, no significant difference was detected between the number of neutrophils that crossed control and ADAM10-deficient endothelial cells. Taken together, these data indicate that loss of endothelial ADAM10 changes the kinetics of neutrophil TEM, but that overall transmigration numbers were unaltered.

### 3.2. ADAM10 Supports Neutrophil Dissociation from the Endothelium

We further investigated the kinetics of TEM and analyzed neutrophil crawling, according to the following definition: Crawling starts after firm adhesion and continues over the complete observation period or is interrupted by probing or diapedesis. According to this definition, we analyzed the DIC movies in more detail using ImageJ plug-in software (see Section 2 for more details) and found that neutrophil crawling time was extended on ADAM10-deficient endothelial cells compared to controls (Figure 3A). Crawling velocity was ultimately not significantly different between the two conditions, but there was a slight trend towards an increase in crawling velocity on ADAM10-deficient cells (Figure 3B). The distance that neutrophils crawled on the endothelial surface until firm adhesion was established, was significantly increased in ADAM10-deficient endothelial cells compared to controls (Figure 3C). Once the neutrophils had completed diapedesis, reaching the basolateral side of the endothelium, the velocity and crawling distance was unaltered (Figure 3D,E). However, the actual diapedesis time, i.e., the time it takes for one neutrophil to fully cross the endothelial monolayer, was significantly increased (Figure 3F). This raised the idea that shedding of ICAM-1 can also take place at the basolateral side of the endothelium to release them into the tissue. To study that, we used Transwell systems [27] where the neutrophils have to be released from the cell layer they pass to end up in the lower compartment. These results showed that neutrophil transmigration across ADAM10-silenced endothelial monolayers was reduced compared to neutrophils that crossed control endothelial monolayers (Figure 3G). Considering these results, we hypothesized that once neutrophils adhere to the endothelium, ADAM10 mediates shedding of the extracellular domain of ICAM-1, allowing diapedesis to be finalized. Therefore, after cleavage of the extracellular domain of ICAM-1, we expected the extracellular domain to be attached to the neutrophil.

To test if ICAM-1 presence on the surface of neutrophils that have crossed the endothelium was increased, we used Transwell assays [25]. Endothelial cells were stimulated overnight with TNFα and complement factor C5a was used as chemoattractant in the lower compartment. Neutrophils that did cross the endothelium and ended in the lower compartment were retrieved and stained for ICAM-1. Additionally, neutrophils that had crossed a bare Transwell filter, i.e., without seeded endothelial cells on top, were also collected and stained for ICAM-1 and VE-cadherin. Flow cytometry analysis of these neutrophils revealed that only neutrophils which crossed an endothelial monolayer showed an increase in intensity for ICAM-1 staining (Figure 4A). This indicates that either the extracellular domain of ICAM-1 is transferred to the neutrophil by shedding or a part of the membrane is transferred onto the neutrophil. No such signal was obtained when the neutrophils crossed a bare filter or when transmigrated neutrophils were analyzed for the presence of endothelial-specific protein VE-cadherin (Figure 4B). Although there is no receptor or ligand present on the neutrophil for VE-cadherin to bind to, this control experiment shows that an antibody as such is not non-specifically transferred onto the transmigrating neutrophil. It also indicates that potential membrane transfer is not occurring at membrane compartments that expressed VE-cadherin. Moreover, when neutrophils crossed ADAM10-deficient endothelium, a decrease in ICAM-1, not VE-cadherin, staining on the transmigrated neutrophil was detected compared to the ones that crossed control endothelial monolayers (Figure 4C), indicating that endothelial ADAM10 contributes to efficient TEM by cleaving ICAM-1. The decrease in ICAM-1 signal was not as large as may have been expected considering the results shown in Figure 4A, although it is difficult to directly compare these data, as Figure 4A displayed fluorescent intensity, and not necessarily number of molecules. One explanation of the limited decrease in ICAM-1 staining upon ADAM10 knockdown may be that ADAM10 knockdown was not 100%. Alternatively, transfer of ICAM-1 containing membrane pieces might also contribute to the increased ICAM-1 signal on transmigrated neutrophils.

### 3.3. ICAM-1 Detection on Transmigrated Neutrophils

To test if not only ICAM-1 but also a piece of the endothelial membrane is transferred onto the transmigrating neutrophil, we transfected endothelial cells with an ICAM-1-GFP construct. The construct was expressed in more than 95% of the cells (Appendix A). We hypothesized that if a piece of membrane is transferred, a green signal i.e., the GFP of ICAM-1, must be detected on the neutrophil, as the GFP tag is cloned to the C-terminus, i.e., the intracellular domain of ICAM-1. Transwells were used as described above and filters were cultured with ICAM-1-GFP-transfected endothelial cells. Neutrophils were allowed to migrate towards C5a that was present in the lower compartment. Image Stream analysis of individual neutrophils showed that 10–15% of the neutrophils that crossed ICAM-1-GFP-positive endothelial monolayers were positive for a green signal (Figure 4D). Many of these neutrophils stained positive for ICAM-1 using an antibody directed to the extracellular domain of ICAM-1, indicating that full length ICAM-1 is transferred to neutrophils during transmigration (Appendix A).

Taken together, we conclude that ADAM10-mediated shedding of the extracellular part of ICAM-1 helps to efficiently cross the endothelium (Appendix A) but clearly is not the major pathway for neutrophils to cross the endothelial barrier. Upon TEM, the extracellular domain of ICAM-1 can travel along with the transmigrated neutrophils, while in some cases ICAM-1-containing membrane parts are also transferred to the neutrophils during TEM.

## 4. Discussion

In this study, we aimed to determine the functional role of endothelial ADAM10 on the shedding of ICAM-1 during neutrophil TEM. The findings presented here reveal that ADAM10, next to shedding of VE-cadherin, is involved in shedding of the extracellular domain of ICAM-1 and in that way contributes to efficient neutrophil TEM. Moreover, our data show that besides transmission of ICAM-1 ectodomain after shedding, also ICAM-1-containing endothelial membrane patches can be transferred to the transmigrating neutrophils.

Previous research by Millan and colleagues demonstrated that endothelial ICAM-1 could travel with migrating T-lymphocytes from the luminal to the basolateral side of the endothelium. This was shown to particularly occur when T-lymphocytes crossed the endothelium in a transcellular manner [8]. This study showed that both ICAM-1 and caveolae translocated to the basolateral side during diapedesis, although it was not clear whether this included only the extracellular part of ICAM-1 or the full length ICAM-1, or whether the mechanism was exclusive to T-cells. Another study showed that LFA-1, the ICAM-1 binding counter receptor on neutrophils, is redistributed to the trailing end of the neutrophil while transmigrating, suggesting that ICAM-1 remains at the apical side of the endothelium [9]. Data from the current study indicates that both full length ICAM-1 and the extracellular ICAM-1 domains can be detected on the neutrophil after transmigration. This suggests potential transfer of endothelial membrane patches to the neutrophil. Indeed, with Western blotting and ELISA analysis, we could confirm ADAM10 is involved in shedding of ICAM-1, releasing the ectodomain.

ADAM10-mediated shedding of ICAM-1 is most likely not the only enzyme essential for efficient TEM. Various studies have implicated ADAM10, but also the related ADAM17, in endothelial barrier function and leukocyte transmigration, by cleaving several leukocyte adhesion molecules (on endothelium and leukocytes) like E-selectin, ICAM-1, VCAM-1 and PECAM-1 or endothelial junctional molecules such as VE-cadherin, JAM-A or claudin 5 [28]. As the total number of neutrophils that finally cross the endothelium was not changed in the flow setup, it suggests that neutrophils can compensate for the loss of endothelial ADAM10, possibly by releasing ADAM10 from the neutrophil [15], or de-activation of β2-integrins as previously described [29]. The time neutrophils took to crawl and adhere on the ADAM10-deficient endothelial surface was significantly increased prior to entering the diapedesis step, compared to control endothelial cells. The increased adhesion is potentially due to increased expression of ICAM-1 on the endothelial surface, indicating that ADAM10 from endothelial cells can regulate the inflammatory response. Additionally, in the absence of ADAM10, VE-cadherin shedding is less efficient as well, and therefore neutrophils may have a harder time crossing these monolayers. Consequently, neutrophils show longer crawling times on the luminal side of the endothelium. We hypothesized that the increased crawling time was due to impaired shedding of ICAM-1 ectodomain from the endothelium in ADAM10-deficient endothelial cells and this may happen at the luminal as well as the basolateral side. Eventually, the lack of ADAM10 may become compensated by ADAM17-mediated cleavage, which may explain why in the end, after 20 min, no differences in number of transmigrated neutrophils was observed anymore. These results are complementary to previous literature on how neutrophils would be released from the endothelium, i.e., by de-activation of the leukocyte integrins LFA-1 and Mac-1 [10], by adding an additional mechanism, endothelial ADAM10 cleavage of ICAM-1. Another role for ADAM10 activity in neutrophil TEM may be the shedding of VE-cadherin. By reducing cell-cell contacts, the barrier is reduced, and this may favor more neutrophils to cross the endothelium.

The relative importance of each of these molecules, i.e., VE-cadherin and ICAM-1, remains difficult to determine, but most likely coordinated shedding of both adhesion molecules and junctional molecules is involved for proper and efficient TEM. Moreover, the exact molecules involved may depend on the cell types, flow conditions and the route of transmigration (e.g., neutrophil TEM may involve slightly different mechanisms or transmigration routes than T-cells) as discussed by Daniel et al. [30] As already mentioned, many of these substrates are not only shed by ADAM10, but also by other ADAM proteases like ADAM17, adding another layer of complexity. ADAM15 has also been implicated in TEM, though this might not involve shedding but rather depend on the disintegrin domain [22].

The complex nature of these mechanisms regulating endothelial permeability and leukocyte transmigration as well as, (small) differences in experimental set-up may explain the partially/seemingly contradictory results reported by different studies. Many studies have used inhibitors, which are notorious for their lack of specificity for either ADAM10 or ADAM17. In some experiments, depending on their set-up, these inhibitors could even affect both endothelial and leukocyte adhesion molecule shedding. Here we show that ADAM10 knockdown confirmed the effects of selective ADAM10 inhibition in endothelial cells regarding neutrophil adhesion and transmigration. Recently, it has been reported that endothelial ADAM10 knockdown regulates the crossing of T-cells by cleavage of VE-cadherin [13,31], while no effect was found on the number of neutrophils that crossed these ADAM10-deficient endothelial cells [31]. However, the timing of analyzing neutrophil transmigration was different. While we started analyzing neutrophil rolling, adhesion and TEM immediately, Reyat et al. [31] first flowed neutrophils over the endothelium for 4 min, then washed away non-adherent cells for at least 9 min, after which recording was started. Importantly, our data showed that the kinetics of neutrophils crossing the endothelium was altered when ADAM10 was inhibited or silenced. We observed reduced neutrophil TEM within the first 5–10 min, but the differences were not observed anymore after 20 min, which is in line with the findings by Reyat et al. Interestingly, in the Transwell system, i.e., under static conditions, we did find a more persistent reduction in neutrophil transmigration, suggesting that besides kinetics, also differences in flow conditions should be considered when analyzing impact of ADAM10-mediated shedding on leukocyte transmigration. This makes perfect sense as shear stress is not only known to affect adhesion molecule expression but has also been shown to enhance ADAM10 activity [32] as well as ADAM17-dependent shedding of L-selectin [33].

ADAM10 cellular localization is regulated by tetraspanins [34], which form specialized microdomains within the cell membrane (tetraspanin-enriched membrane rafts). Interestingly, upon clustering/leukocyte binding, ICAM-1 has been reported to be recruited to tetraspanin microdomains [35,36] as well as caveolin-rich lipid rafts [37]. It remains to be determined if the ADAM10-ICAM-1 interaction depends on and is regulated by the presence of such membrane raft localization.

Taken together, for neutrophils to efficiently cross the endothelial layer, ICAM-1 can be cleaved from the endothelial surface by ADAM10. We emphasize that this process is probably not crucial for neutrophils to cross, but occurs, possibly for optimal efficiency, and most likely works in conjunction with the leukocyte integrin de-activation pathway to allow extravasation of neutrophils through the endothelium [10]. Additionally, we found that patches of ICAM-1-rich membranes can be transferred from the endothelium onto the transmigrating neutrophil, although we wish to stress that this happens for a small percentage of transmigrated neutrophils. Our findings raise additional questions. For instance, how is ADAM10 activated during neutrophil diapedesis? Is ICAM-1 shedding dependent on neutrophil TEM or does it occur to the same extent during neutrophil crawling? In addition to that, what does the membrane transfer mean in terms of TEM for other leukocyte subsets. These highly relevant questions may lead the way to future experiments and deserve a clear answer to fully understand the role of ADAM10 in neutrophil TEM.

## Figures and Tables

**Figure 1 cells-10-00232-f001:**
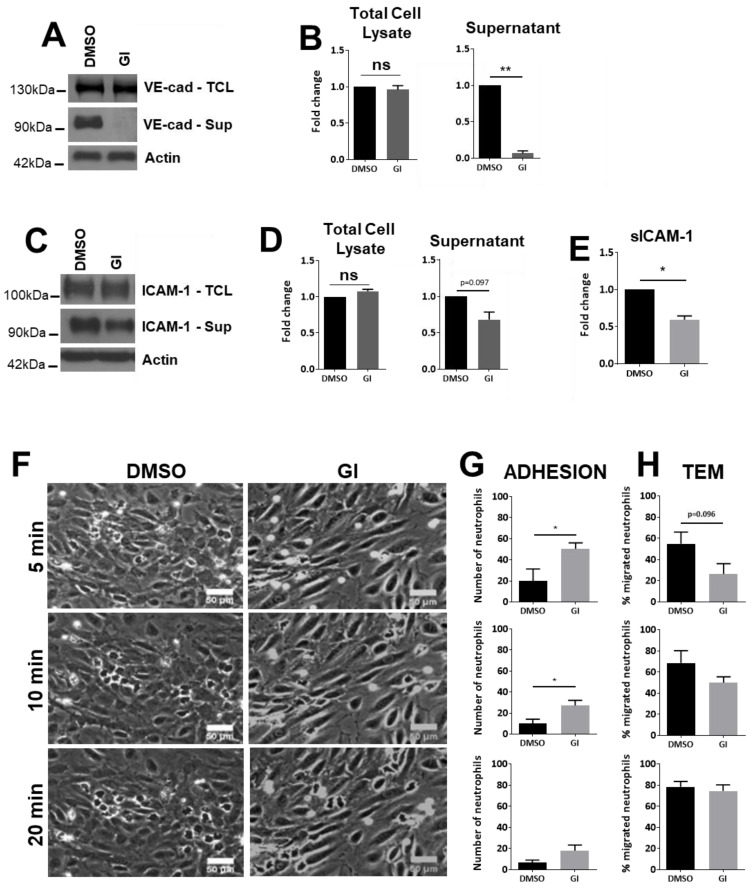
ADAM10-regulated ICAM-1 Protein expression levels in HUVECs. (**A**) HUVECs cultured and treated as indicated (10 ng/mL TNFα) and analyzed on the expression of VE-cadherin by Western blotting. GI: ADAM10 inhibitor. Upper panel shows total cell lysate (TCL), middle one extracellular domain of VE-cadherin present in the supernatant of cultured ECs and lower panel shows actin as protein loading control for TCL. (**B**) Quantification of VE-cadherin expression in TCL and the extracellular domain of VE-cadherin present in the supernatant. (**C**) HUVECs were treated with inhibitors as indicated and ICAM-1 levels were analyzed in total cell lysates and endothelial cell supernatant by Western blotting, as described in A. Note that several ICAM-1 isoforms can exist ([26]. However, we did not find additional bands. (**D**) Same as under B but then for ICAM-1, TCL on left and extracellular ICAM-1 on the right. (**E**) Quantification of soluble ICAM-1 fragment (sICAM-1) in the supernatant measured by ELISA. (**F**) HUVECs were cultured in flow chambers and treated as indicated, followed by perfusion with neutrophils for indicated time points. Bar, 50 µm. (**G**) Quantification of number of neutrophils that adhered to TNFα-treated ECs, displayed for different time points as indicated under F. (**H**) Quantification of number of neutrophils that crossed TNFα-treated ECs (TEM: transendothelial migration), displayed for different time points as indicated under F. Data are mean of at least three independent experiments. * *p* < 0.05; ** *p* < 0.01.

**Figure 2 cells-10-00232-f002:**
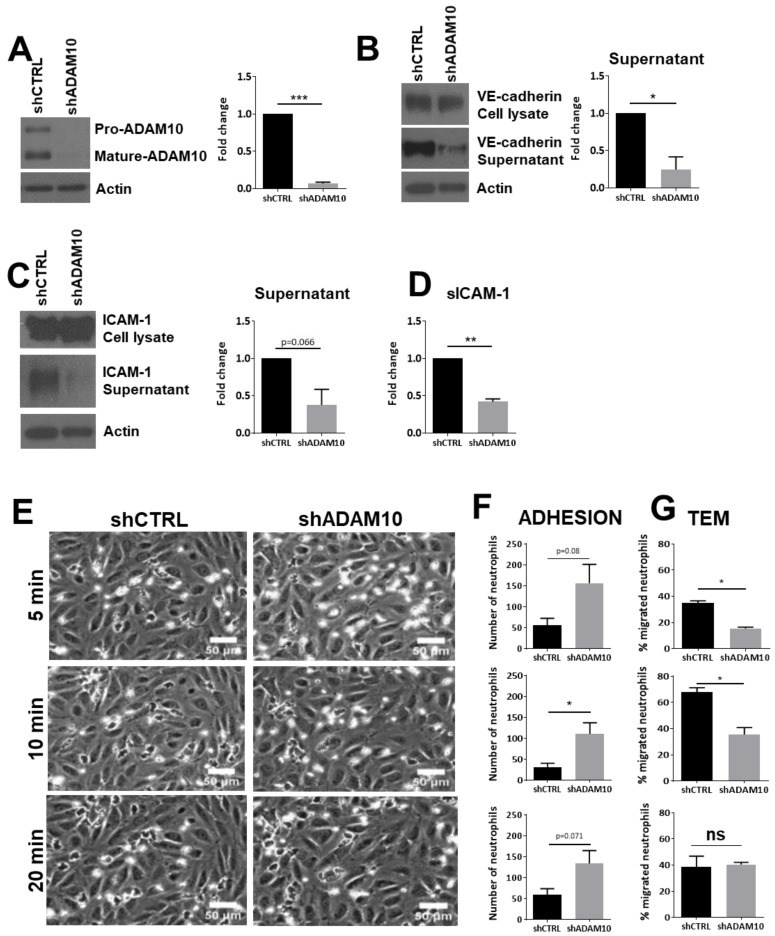
Leukocyte transendothelial migration upon ADAM10 inhibition. (**A**) Cultured HUVECs were treated with shCTRL or shADAM10, checked by Western blotting. Actin was used as loading control. Quantification shows significant reduction of mature ADAM10 in ECs upon shRNA treatment. (**B**) ADAM10-deficient ECs show reduced shedding of VE-cadherin extracellular domain. Western blotting shows VE-cadherin expression in total cell lysate (upper panel), VE-cadherin extracellular domain in supernatant (middle panel) and actin as a loading control in lower panel. Quantification on the right shows significant reduction of extracellular VE-cadherin domain. (**C**) ADAM10-deficient ECs show reduced shedding of ICAM-1 extracellular domain, analyzed by Western blotting. Quantification on the right. (**D**) Soluble ICAM-1 levels were measured using ELISA. Reduced sICAM-1 in supernatant is measured in ADAM10-deficient ECs. (**E**) HUVECs were cultured in flow chambers and treated as indicated, followed by perfusion with neutrophils for indicated time points. Bar, 50 µm. (**F**) Quantification of number of neutrophils that adhered to TNFα-treated ECs, displayed for different time points as indicated under E. (**G**) Quantification of number of neutrophils that crossed TNFα-treated ECs, displayed for different time points as indicated under E. Data are mean of at least three independent experiments. * *p* < 0.05; ** *p* < 0.01; *** *p* < 0.001.

**Figure 3 cells-10-00232-f003:**
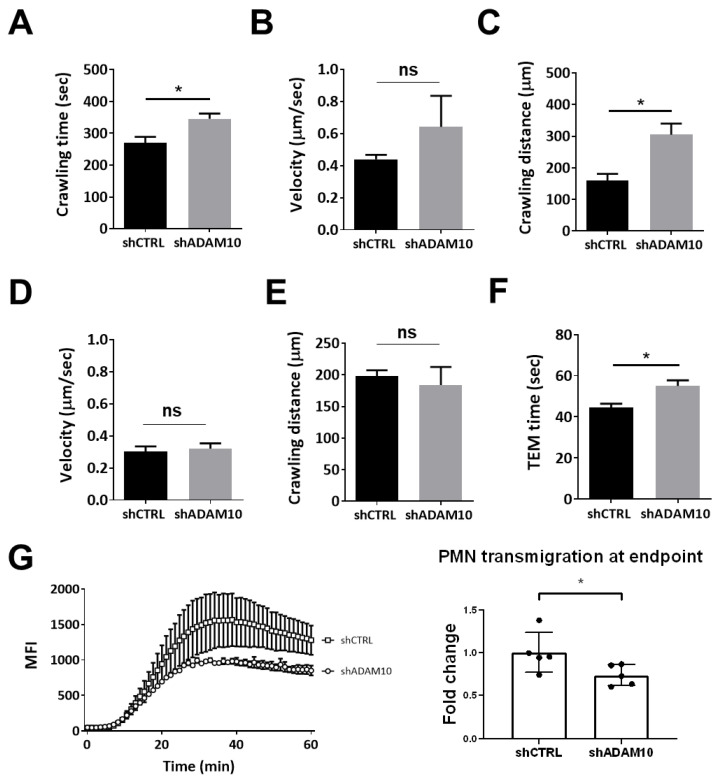
Adherent and crawling neutrophils on apical surface of TNFα-treated endothelium. (**A**) Crawling time (in seconds) of neutrophils from adhesion to diapedesis was determined for neutrophils that crossed either shCTRL or shADAM10-treated EC monolayers. (**B**) Velocity (µm/s) and (**C**) distance (µm) are determined on neutrophils that were on top (apical) of the endothelium that was silenced for ADAM10 (shADAM10) or treated with shCTRL. (**D**) Velocity (µm/s) and (**E**) distance (µm) are determined on neutrophils that were underneath (basolateral) of the endothelium that was silenced for ADAM10 (shADAM10) or treated with shCTRL. (**F**) Time (in seconds) of actual diapedesis measured for neutrophils that crossed either shCTRL or shADMA10-treated endothelial cell monolayers. Data are mean of at least three independent experiments. * *p* < 0.05. (**G**) Transwell system is used to allow calcein-labeled neutrophils to cross an endothelial monolayer that was cultured on a FN-coated filter with 3 µm pore size including fluorescent block. Increase in fluorescence was detected in real-time using a fluorimeter and showed increased neutrophil migration towards C5a. Squares represent shCTRL-ECs and open circles represent shADAM10-ECs. Graph on the right shows quantification after 20 min. Data are mean of at least three independent experiments. * *p* < 0.05.

**Figure 4 cells-10-00232-f004:**
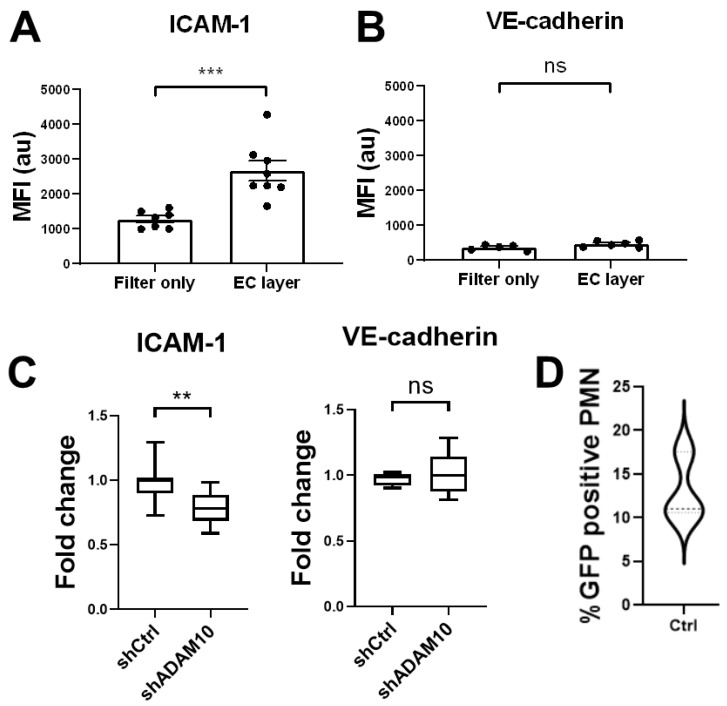
Transmigration of neutrophil through endothelial monolayers. (**A**) Detection of ICAM-1 using antibodies that exclusively recognize extracellular part of ICAM-1 indicate that only neutrophils that crossed the endothelium (labeled as EC layer) show an increase of the presence of ICAM-1 on their surface, not when they crossed a bare filter (labeled as Filter only). (**B**) No increase in VE-cadherin is detected when using anti-VE-cadherin antibodies that detect the extracellular domain of VE-cadherin. (**C**) Silencing of EC-ADAM10 reduced the amount of ICAM-1 on the surface of the neutrophils compared to crossing shCTRL-treated EC monolayers, once they have crossed the endothelial monolayer. As a negative control, no change in VE-cadherin detection was measured (graph on right). Data are mean of at least three independent experiments. ** *p* < 0.01; *** *p* < 0.001. (**D**) HUVECs were transfected with ICAM-1-GFP, subsequently treated with TNF and neutrophils were allowed to cross ICAM-1-GFP-HUVECs in Transwell system. 10–15% of neutrophils that crossed transfected EC showed green signal, indicating membrane transfer during TEM. Experiment is carried out three times in duplicate.

## Data Availability

No new data were created or analyzed in this study. Data sharing is not applicable to this article.

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
