# Peer review of "ADAM10-Mediated Cleavage of ICAM-1 Is Involved in Neutrophil Transendothelial Migration"

_cells, 2021, doi:10.3390/cells10020232_

Round 1

Reviewer 1 Report

Major request:

The authors should clearly delineate whether any their results is a reproduction of data published before and which data are new. Which of the ADAM-10 knock-down or inhibition effects can be attributed to shedding of VE-cadherin and which effects can be attributed to shedding of ICAM-1? Also, the authors should demonstrate that only the extracellular domain of ICAM-1 but not the intracellular domain of ICAM-1 is carried on by the neutrophils after diapedesis (comment: The GFP-ICAM-1-mCherry construct would be suited for this request).

Minor requests

Lines 74 – 77, Antibodies and reagents
Please comment on ICAM-1 splice isoforms recognized by the anti-ICAM-1 antibodies used. The discussion should  elaborate the possible impact of non-detected ICAM-1 isoforms on the results of this study.

Lines 236 - 239
Not the efficiency but rather the time kinetics of neutrophil TEM was changed upon loss of endothelial ADAM10

Lines 260 – 262, Fig 3C
Please explain the definition of crawling in the manuscript. As a comment: The general definition is as follows. Under flow, neutrophils tether and roll until they undergo firm adhesion to the endothelial surface. Crawling starts only after firm adhesion and continues over the complete observation period or is interrupted by probing or diapedesis.

Lines 266 – 269
ADAM10 mediated shedding of ICAM-1 while engaged to LFA-1 or MAC-1 on the neutrophil. Thrilling hypothesis. However, what promotes ICAM-1 shedding in the absence of neutrophils (Fig 2)? How is ADAM10 activated during neutrophil diapedesis? Is ICAM-1 shedding dependent on neutrophil TEM or does it occur to the same extend during neutrophil crawling? Is increased ICAM-1 shedding observed in the presence of neutrophils prior to diapedesis (Western Blot of TCL with detection of the ICAM-1 cytoplasmic tail only)?

Lines 280 – 282
According the hypothesis before, bound extracellular domain of ICAM-1 should be found on the neutrophil surfaces. But this is different from “increased expression”. Please clarify.

Lines 285 – 287 and Fig 4
A differentiation between bound ICAM-1 on the neutrophil surface and increased ICAM-1 expression by the neutrophils upon diapedesis must be presented (qPCR on neutrophil cDNA after TEM?). – see also below comment to line 317

Lines 28- 289, Fig 4B
Please check language. In principle, the wording here is correct -  for FACS analysis the neutrophils are incubated with an antibody. But here, this wording is misleading. Maybe: ….transmigrated neutrophils were “stained” for VE-cadherin….

Lines 307 – 311
Rephrase sentence. Object missing.

Line 317 – Figure 5 B
The finding that transmigrated neutrophils take along the GFP tagged extracellular domain of endothelial ICAM-1 answers in part my request above for lines 285 – 287. Still, increased expression of ICAM-1 in neutrophils upon diapedesis should be excluded.

Lines 318 – 320, fig 5C
The absence of mCherry on neutrophil surface after diapedesis across GFP-ICAM-1-mCherry HUVECs should be shown.

Fig 5C
Please explain the absence of ICAM-1 antibody signal on neutrophils that migrated across YFP-CAAX or GFP transfected HUVECs.

Author Response

We wish to thank reviewer 1 for his/her thoughtful comments. We have addressed the comments of the reviewer to the best of our abilities and listed them in a point-by-point reply below.

Reviewer 1
Major request:
The authors should clearly delineate whether any their results is a reproduction of data published before and which data are new. Which of the ADAM-10 knock-down or inhibition effects can be attributed to shedding of VE-cadherin and which effects can be attributed to shedding of ICAM-1? Also, the authors should demonstrate that only the extracellular domain of ICAM-1 but not the intracellular domain of ICAM-1 is carried on by the neutrophils after diapedesis (comment: The GFP-ICAM-1-mCherry construct would be suited for this request).

The reviewer raises a valid point. As a major player of vascular permeability and the main component of adherens junctions of endothelial cells, VE-cadherin is essential for the control of vascular integrity. Several research groups, including ours, have shown that blocking VE-cadherin, by interfering antibodies, or silencing, promotes the transmigration of leukocytes across endothelial monolayers. This was also shown for in vivo to happen. Therefore, by shedding VE-cadherin, the integrity of the endothelial monolayer is hampered, and permeability is increased. As a result, one would expect that shedding of the extracellular part of VE-cadherin would increase the number of leukocytes that cross the endothelium. However, work by the Vestweber lab showed that locking VE-cadherin to the actin cytoskeleton, by introducing a VE-cadherin-alpha-catenin chimera prevents leukocytes transendothelial migration (TEM) (Schulte et al., EMBO J, 2011). And as VE-cadherin at the extracellular part of this construct is no different than the part of the wildtype VE-cadherin, it is expected that ADAMs would still be able to shed this part of VE-cadherin. The fact that TEM is blocked indicates that VE-cadherin shedding is not directly controlling leukocyte TEM. We have added this option to the discussion part of the revised paper.
In figure 4 and 5, we have used a double tagged ICAM-1 construct. This construct has a GFP tag at the N terminus, meaning the extracellular part and a mCherry construct at the intracellular part, the C-terminus. Our data show that green signals can be detected on neutrophils that have crossed the endothelial cells when transfected with this construct. Moreover, we have included real-time video and stills from this video that show the presence of only a green signal on the migrating neutrophil, but no red signal. These data indicate that part of the GFP-tagged ICAM-1 can be detected on neutrophils that have crossed the endothelium. We have also used membrane tagged GFP, as a control. In these control experiments iit became clear that there is a possibility that some membrane is transferred as well. Thus, we have toned down and adjusted our conclusions to these findings.

Minor requests
• Lines 74 – 77, Antibodies and reagents
• Please comment on ICAM-1 splice isoforms recognized by the anti-ICAM-1 antibodies used. The discussion should elaborate the possible impact of non-detected ICAM-1 isoforms on the results of this study.

We agree on that with the reviewer. We have added a comment and an extra reference on this in the result section non page 7.

• Lines 236 - 239: Not the efficiency but rather the time kinetics of neutrophil TEM was changed upon loss of endothelial ADAM10
We have changed this in the text.

• Lines 260 – 262, Fig 3C: Please explain the definition of crawling in the manuscript. As a comment: The general definition is as follows. Under flow, neutrophils tether and roll until they undergo firm adhesion to the endothelial surface. Crawling starts only after firm adhesion and continues over the complete observation period or is interrupted by probing or diapedesis.
We thank the reviewer for his/her help here and have added the definition for crawling to the revised Ms.

• Lines 266 – 269: ADAM10 mediated shedding of ICAM-1 while engaged to LFA-1 or MAC-1 on the neutrophil. Thrilling hypothesis. However, what promotes ICAM-1 shedding in the absence of neutrophils (Fig 2)? How is ADAM10 activated during neutrophil diapedesis ? Is ICAM-1 shedding dependent on neutrophil TEM or does it occur to the same extend during neutrophil crawling? Is increased ICAM-1 shedding observed in the presence of neutrophils prior to diapedesis (Western Blot of TCL with detection of the ICAM-1 cytoplasmic tail only)?

The reviewer raises some remarkably interesting points. All questions that are difficult to address. We have taken the opportunity to put these questions into the discussion section, for future perspective and direction. Unfortunately, time is the limiting factor for us and keeps us from further exploring these challenging ideas at this moment.

• Lines 280 – 282: According the hypothesis before, bound extracellular domain of ICAM-1 should be found on the neutrophil surfaces. But this is different from “increased expression”. Please clarify.

The reviewer is right, and this was clearly a miswriting. We have changed this in the revised Ms.

• Lines 285 – 287 and Fig 4: A differentiation between bound ICAM-1 on the neutrophil surface and increased ICAM-1 expression by the neutrophils upon diapedesis must be presented (qPCR on neutrophil cDNA after TEM?). – see also below comment to line 317

In line with the previous comment, we do not see any increase in expression of ICAM-1 on the surface of the neutrophil and think that the increased presence of ICAM-1 on the neutrophil is solely due to the transfer of shed ICAM-1 from the endothelium. We will make this clearer in the result section of the revised Ms.

• Lines 28- 289, Fig 4B: Please check language. In principle, the wording here is correct - for FACS analysis the neutrophils are incubated with an antibody. But here, this wording is misleading. Maybe: ….transmigrated neutrophils were “stained” for VE-cadherin….

We have rephrased these sentences in the result section at page 11.

• Lines 307 – 311: Rephrase sentence. Object missing.
Done.

• Line 317 – Figure 5 B: The finding that transmigrated neutrophils take along the GFP tagged extracellular domain of endothelial ICAM-1 answers in part my request above for lines 285 – 287. Still, increased expression of ICAM-1 in neutrophils upon diapedesis should be excluded.

As stated above, in figure 4 and 5, we have used a double tagged ICAM-1 construct. This construct has a GFP tag at the N terminus, meaning the extracellular part and a mCherry construct at the intracellular part, the C-terminus. Our data show that green signals can be detected on neutrophils that have crossed the endothelial cells when transfected with this construct. Moreover, we have included real-time video and stills from this video that show the presence of only a green signal on the migrating neutrophil, but no red signal. These data indicate that part of the GFP-tagged ICAM-1 can be detected on neutrophils that have crossed the endothelium.

In the Transwell assay, figure 4, we have stained neutrophils with ICAM-1 antibodies and analyse these with flow cytometry. Here we could show that ICAM-1 levels were not increased on neutrophils when they were left untreated or when they have crossed a bare filter towards the same chemokine C5a as was used in the Transwell assays when the filter was covered with a monolayer of endothelial cells. In the latter condition, we found an increase in the detection of ICAM-1. To us, this indicates that the ICAM-1 was coming from the endothelium, and this was confirmed by the experiments we performed in figure 5, using the double tagged ICAM-1 construct.
We have added these remarks to the revised manuscript.

Lines 318 – 320, fig 5C: The absence of mCherry on neutrophil surface after diapedesis across GFP-ICAM-1-mCherry HUVECs should be shown.

We have re-analysed our Image Stream data and included the red channel now in the analysis. We are happy to say that no specific signal was detected here. We analysed the neutrophils that crossed GFP-ICAM-1-mCherry transfected endothelial cells. The analysis showed that neutrophils were positive for the GFP signal and stained positive when incubated with an IAM-1 antibody. However, the red channel did not show any signal, indicating that the mCherry part of ICAM-1 was left behind on the endothelium. We have added these new data to the manuscript as Figure S1C and adjusted the text accordingly in the revised version.

Fig 5C: Please explain the absence of ICAM-1 antibody signal on neutrophils that migrated across YFP-CAAX or GFP transfected HUVECs.

For these experiments, we did see antibody staining on CAAX positive events, however, not as frequent as with the ICAM-1-GFP samples. This is probably due that the CAAX cells may also present endogenous ICAM-1 but not as much as ICAM-1-GFP. Therefore, the signal as such is lower in intensity. We have increased the intensity a bit more for this specific sample and could detect an ICAM-1 antibody signal. We have included these extra data for the reviewer’s purpose below (R1_Figure 1). We have added these remarks to the result section of the revised manuscript on page 12.

Reviewer 2 Report

The authors present data suggesting that ADAM10 is essential for transendothelial migration (TEM) of neutrophils across HUVEC monolayers. The study is generally well conceived and executed, and discussion of results is for the most part appropriate. Some minor changes should be made as outlined below:

Comments:

-In the methods description, there appears to be no comment on the degree of purity from the neutrophil isolation. Please indicate the method of determination of cell purity and degree of purity from this prep.

-Line 206, the authors claim that adhesion is higher and TEM is lower at 5 and 10 minutes. This is true for 5 minutes, but only adhesion is lower at 5 min. There is no significant change in TEM indicated in Figure 1G at 10 min.

-More discussion of the curious result that after 20 minutes, TEM in the ADAM10 inhibitor and shADAM10 groups were equivalent to untreated cells is warranted. Are there other mechanisms/metalloproteinases that may be involved and explain this result?

Author Response

Response in the attached PDF file.

Reviewer 3 Report

The study ‘ADAM10-mediated cleavage of ICAM-1 is required for neutrophil dissociation from the endothelial surface during the final diapedesis step’ by Morsing et.al., investigates the potential role of proteinase cleavage in promoting transendothelial migration (TEM) with the stated focus on the ultimate cell-cell de-adhesion step of diapedesis. Overall, the authors use either pharmacologic or endothelial shRNA knockdown (KD) of ADAM10 to assess ICAM-1 and VE-cadherin shedding (by Western blot and ELISA) and imaging studies to monitor neutrophil adhesion and TEM dynamics. The results of these studies seem to suggest a modest and transient effect of ADAM10 on neutrophil diapedesis dynamic. Transwell migration experiments were then used to assess transfer of endothelial ICAM-1 to neutrophil membranes.

            At face value this study seems potentially interesting, but a number of conceptual and technical issues emerge that significantly undermine the study. First, the title is incongruent with the findings. The statement that ADAM-10 is ‘required’ for neutrophil dissociation is an overstatement as the authors at least partially acknowledge in the final sentence of the discussion stating: ‘We emphasize that this route of transmigration is probably not crucial for neutrophils to cross, but is important for optimal efficiency’. In deed, the main supporting experiments in Fig.1 and 2 show modest and transient effects (discussed further below). Moreover, a previous study (Ref.22, Fig.1) already conducted a nearly identical experiment as the one described here in Fig.2 and show no effects of ADAM10 KD on neutrophil adhesion or transmigration across similarly activated HUVEC. A second major issue is a number of confounding variables that are outlined below. Finally, the overall interpretation of the main results seems fundamentally flawed (see ‘interpretation’ below).

Figure 1 assesses responses of HUVEC to pharmacologic inhibition of ADAM10 and finds that both ICAM-1 and VE-cadherin shedding are reduced, thereby preserving higher levels of these molecules that are functionally intact at the cell surface and junctions, respectively. In reality, the effects on VEC (which are well reported in refs 13 and 22) are by comparison much stronger (see further point below).

Under these conditions neutrophils are stated to exhibit increased adhesion but reduced TEM. However, the images to support this in F are ambiguous and show yellow and red arrow (not mentioned in the legend) that I infer might be intended to mark adherent and transmigrated cells. These images are not clear or convincing. Moreover, the methods state the immunofluorescence analysis was used for determine and quantify TEM (only after 20 min). No fluorescence images are provided in this regard and it remains unclear precisely how these results were obtained. Moreover, the differences in adhesion and TEM were stated to be evident at 5 and 10 min, but lost at 20 min. In fact, there were only significant differences at 5 min.

Confounding variable: GI254023X (GI) inhibits ADAM10 with an IC50 of 5.3 nM for ADAM17 with an IC50 of 541 nM. The authors use GI at 5 micromolar, which will inhibits ADAM17 by ~90%. Thus, the role of AMAD10 versus ADAM17 is ambiguous in these experiments. Indeed, the finding that VE-cadherin shedding is reduced by ~95% with GI and only ~75% by EC shRNA KD of ADAM10 (Fig.2 below) supports this idea.

Confounding variable: As reported previously, and substantiated here, ADAM10 seems to play much more profound role (compared with ICAM-1) in promoting shedding of the pivotal barrier protein VE-Cadherin that stabilizes EC junctions and limits diapedesis. Previous work (refs 13, 22) established lymphocyte diapedesis was reduced by ADAM10 blockade in a manner that depended on VE-cadherin stabilization. The authors neither discuss how ADAM10-related VE-cad stabilization might effect their results (see further below) nor do anything to control for or interrogate this intuitive possibility.

Figure 2 largely recapitulates Fig.1 using knockdown of ADAM10 rather that an inhibitor. Critically, again only modest and very short-term effects are seems that disappear by 20 min. In physiologic terms, where acute inflammatory reactions develop quickly but persist for hours-days the relevance of this brief effect seems unclear. Fig.2E-G present the same concern regarding imaging and analysis as raised in Fig.1F-H. However, this figure raises a further point of confusion. It seems odd that the percentage transmigrated cells in the control goes from ~35% at 5 min, up to ~70% at 10 min and back down to ~35% at 20 min. Did these cells reverse transmigrate?

Figure 3 shows further analyses related to experiments in Fig. 2. These indicate that ADAM10 knockdown on EC lead neutrophils to laterally migrate for longer times and for greater distances (e.g., ~300 um versus ~150 um for control) before TEM. They also show a clear trend (though not statistical significant) toward an increased lateral migration velocities. In addition once initiated the entire duration of the final TEM event was also slightly increased, but the velocity and lateral migration distance of sub-endothelial neutrophils was not different in control and ADAM10 KD. Overall, there is insufficient information on how all of these steps were determined. How, was apical versus sub-endothelial and the ‘TEM’ event identified? What were the markers and criterion for each? Representative images? Again, methods refer to immunofluorescence for these analyses: What stains? How imaged?

In addition to these technical concerns is a fundamental issue with interpretation. It is universally agree that ICAM-1 plays its most critical function in initial firm adhesion, spreading and apical lateral migration. Some have suggested that ICAM-1 is less important during late steps in diapedesis in favor of other adhesion molecules, such as PECAM-1. While I do not personally espouse this latter notion, it is clear that ICAM-1 is critical in the early spreading and traversing of leukocytes over the apical surface. It is also well established that trailing edge de-adhesion is as critical as the formation of new leading edge adhesions to drive effective lateral migration. Numerous studies have investigated mechanism for de-adhesion processes and consistently find that when these are interrupted migration velocities and distances are reduced. Specifically, new adhesion form, but cells are stuck because trailing edge fails to release efficiently. The central theme and conclusion presented in this study is that ADAM10 plays are role in releasing ICAM-1 from the endothelium in ways that help release the trailing edge during migration. If this was truly important, ADAM10 inhibition or KD would cause reduction in lateral migration that would be evident immediately following firm arrest spreading and polarization, when the neutrophils first begin to move. But this is not what the authors find. Rather, they report the opposite; in the absence of ADAM10 the neutrophils cover twice the distance, ~300 um (or ~30 cell lengths) with velocities (tending toward) 30% faster. Thus the experiments simply are inconsistent with a critical role for ADAM10 in regulating ICAM-1-neutrophil de-adhesion. Rather this study seems to show that in absence of ADAM10 it takes neutrophils longer distance/time to explore the EC surface and slightly more time to execute the actual TEM step. This is much more inline with aforementioned defect in VEC shedding, which has previously been linked to reduced T cell TEM due to less efficient junctional opening (refs 13, 22).

Figures 4 & 5 attempt to establish that endothelial ICAM-1 is transferred to neutrophils during diapedesis and that this is supported by ADAM10. Figure 4 shows that ICAM-1 is indeed increased on neutrophils following diapedesis across endothelium but naked transwells consistent with potential direct transfer of endothelial ICAM-1. This is not surprising and is an already well established finding, which has been often explicitly linked to membrane transfer that is depends on high-density ICAM-1/integrin adhesion clusters. It also shows that ADAM10 KD has a minute effect on this transfer, indicating that it is, in fact, mostly due to membrane transfer of intact ICAM-1 rather than cleavage and transfer of extracellular domain. Figure 5 presents a novel ICAM-1 molecule that is differentially labeled with N and C terminal fluorescent proteins. While in theory this might have made for an interesting tool, the experiment is confounded by the confusing use of fluorescent antibodies that overlap with fluorescence signals of the FPs. In Figure 5B, YFP-CAAX is an insufficient control. Since membrane transfer is known to occur in an ICAM-1-mediated fashion, the membrane that gets transferred is always densely enriched in ICAM-1 (or in this case ICAM-1-GFP), though some YFP-CAAX will be incorporated in any membrane that is transferred it would not be enriched would be expected exhibit relatively poor signal for an equal amount of membrane. Images and C and D are ambiguous.

Author Response

Response in the attached PDF file.

Round 2

Reviewer 1 Report

All comments and requests have been carefully addressed. Only the request for qPCR to exclude ICAM-1 upregulation on transmigrated neutrophils is missing. Though, since the manuscript improved significantly, it can be recommended for publication.

Author Response

We wish to thank Reviewer 1 for his/her thoughtful comment, and we were delighted to read that reviewer 1 was satisfied with the changes we have made to our manuscript.

Reviewer 3 Report

The authors have made some improvements, including acknowledgement of some of the limitations of the study. These include a revised title and a few alterations in the results/discussion that acknowledge that the results seem to support a non-essential role for ADMAM10-mediated ICAM-1 shedding that might influence efficiency. They have also recognized the ADAM10/ADAM17 specificity issue regarding the GI inhibitor and acknowledged the transient nature of the effects and added kinetic studies in Fig.3 to provide better characterization of this aspect. Unfortunately, several issues that were raised by multiple reviewers remain inadequately addressed. The two most most problematic are described below.

Issue 1 – Ambiguity about impact of ICAM1 vs VEC shedding: ADAM10 is well known as a VEC sheddase. This was confirmed here while also showing it also sheds ICAM1. The authors have previously advocated that shedding of VEC by ADAM10 facilitates leukocyte diapedesis (e.g., Daniel, JII, 2013). Yet here on the basis of a single study (Schulte, 2013 EMBO; btw, cited twice as Ref#13 & 28) the authors argue that the effects of ADAM10 on VEC should be discounted. This is problematic both because it dismisses a substantial body of contradictory literature and because the argument itself is flawed.

First, the argument: In the discussion (lines 417-421) the authors refer to work by the Vestweber group who engineered an artificial VEC with alpha catenin covalently fused to its C-terminus and showed it stabilized EC junctions and reduced both permeability and neutrophil diapedesis (Schulte, 2013 EMBO). The author of this MS suggest that since the engineered VEC retained the wild type ADAM10 cleavage site but still reduced neutrophil diapedesis, that ADAM10 activity towards VEC must not be involved in neutrophil TEM. The logic then follows that the authors observed effects of ADAM10 inhibition/KD in the current study must be a consequence of alteration of IC1 shedding only. Unfortunately, this argument has several issues. First, the study by Schulte et.al is mostly in vivo and under largely non-comparable conditions. Second, the study by Schulte et.al., was not designed, nor did not attempt, to address shedding in anyway. Finally, the basic mechanism for the barrier stabilization by the VEC-alpha-catenin construct was shown to be stabilization of VEC-actin association, which results increased junctional clustering and mechanical coupling. Both of these effects could impact accessibility and/or conformation of ADAM10 cleavage sites on VEC and contribute to its barrier stabilizing phenotype of this construct. Indeed, both conformational and steric alterations are known mechanisms of regulating proteolytic susceptibility in general.

Second, the broader literature: In addressing this specific issue of defining roles for IC1 vs VEC shedding the authors ignore a significant body of literature that supports the contribution of ADAM10 in destabilizing AJ via shedding of VEC (as well as JAM-1 and claudin5) and promoting endothelial permeability and diapedesis. As note above, the authors themselves have advocated this idea in review (e.g., Daniel, 2013 JII) without citing such work herein. The authors have cited some of relevant literature in the current MS (refs#12-15), but only in the context of a general introduction and without comment on the contradictory findings related to VEC shedding. Moreover, there are many others uncited papers that show key functional roles of endothelial ADAM10 in barrier function and cellular trafficking. Below we briefly highlight just a few examples.

Speck et.al. (Speck, 2015 PMID: 25926412) show that an athero-protective diet lowers EC ADAM10 and ADAM17 and reduces shedding of VEC, JAM1, which improved EC barrier function and reduced macrophage diapedesis. Flemming et.al. (Flemming, 2015 PMID: 25975259) show that VEC shedding and barrier dysfunction in sepsis in mediated by ADAM10 and inhibited by GI254023X. Kabacik et.al., (Kabacik, 2017 PMID: 25975259) and Kouam et.al. (Kouam, 2019 PMID: 28600292) show that ionizing radiation induced VEC and claudin5 cleavage and increased EC permeability and transmigration of tumor cells in a GI254023X inhibitable.

Finally a study by Reyat et.al. (Reyat, 2017 PMID: 28600292) which was highly similar to the current study shows that siRNA knockdown of ADAM10 in TNF-activated HUVEC reduced diapedesis of T cells (by ~50%) but not did not effect neutrophils. Thus it was shown in this study that EC ADAM10-dependent shedding of both IC1 and VEC are both dispensable for neutrophil transmigration. Of note, several experiments in this study show that neutrophil proteases promote EC junction destabilization and even rescue T cell diapedesis when EC ADAM10 is knocked down.

Altogether, the presentation of findings in the current MS is remains poorly integrated with the current literature around VEC shedding in highly relevant contexts.

Issue 2 – Ambiguity regarding mode of ICAM1 transfer to neutrophils Fig4/5: Experiments in described in Figs 4 and 5 attempts to address the idea that endothelial IC1 transfers to transferred neutrophils through ADAM10-mediated shedding. Reviews 1 and 3 both raised significant concern regarding controls, experimental design and interpretation. The authors have added some new data, but unfortunately these have not addressed the core concerns.

Figure 4: As stated in the original comments the staining in Fig.4 does not distinguish between ICAM-1 that is transferred through shedding versus membrane transfer. Fig.4A shows that ICAM1 staining signal on neutrophils increases 2-fold following migration across endothelium indicating transfer (via ambiguous mechanism). Fig.4C shows that only a minor fraction (~15%) of that transferred ICAM-1 signal is lost when EC ADAM10 is knocked down. This demonstrates that the majority (~85%) of transferred ICAM-1 occurs independently of ADAM10 suggesting a dominant contribution of membrane transfer. The authors have added VEC staining in Fig4B and C, which show that VEC is not found on the neutrophils. This is a trivial result since (whether shed or not) VEC has no cognate receptor on neutrophils to bind it. These latter results with VEC therefore do not add anything to the study.

Figure 5: The new construct (Fig.5A) places an N-terminal GFP immediately preceding the domain 1 of ICAM-1, which is a key site for binding by neutrophil LFA1 and for ICAM1 homotypic dimerization on the cell surface. The authors have not conducted any analysis to confirm whether it retains normal expression, surface distribution and function. They also do not describe the sequence and linker for their construct. It should be noted that this issue was not raised in the original review, but nonetheless, it now has come to my attention and more information in the regard needs to be provided.

Regarding Fig5B-D, none of the original concerns (many of which were also shared and raised by reviewer 1) were addressed. As I discussed in the first review, YFP-CAAX can serve as a qualitative tool to indicate membrane transfer. Indeed, with this maker the authors show in several different experiments that membrane is transferred. However, arguments based on quantitation of fluorescence signal of YFP-CAAX vs GFP-IC1 (or anti-IC1 Ab) are not appropriate. First, at the very minimum one would need to establish the baseline fluorescent signals in the EC following transfection (both % transfection efficiency and expression levels/fluorescence intensity range in the positive transfectants). No such analysis was made. Therefore, there is no basis for which to evaluate the meaning of the relative YFP-CAAX vs GFP-IC1 signal intensities transferred to the neutrophils.

More importantly, YFP-CAAX is a generic membrane marker that evenly distributes without bias, while ICAM-1 (or its fluorescent equivalent) gets highly clustered at sites of leukocyte contact (trailing edge/uropod in particular) via binding of leukocyte integrins; an observation the is very well established in the literature. Thus, even when membrane is transferred, it would be anticipated that the GFP-ICAM1 signal would be relatively greater because it is actively concentrated via adhesion. A much more appropriate and useful comparison might have been GFP-IC1 versus IC1-GFP (i.e., cytoplasmic IC1 signal). If initial expression levels were comparable, the differential signal transfer would have very nicely represented the relative membrane transfer versus shedding-dependent transfer. Quantitative FACs with and without ADAM10 inhibition or knockdown is such a set up could clarify these issues effectively. Another, possible approach would be similar quantitative experiments using GFP-ICAM1 and ICAM1-mcherry co-transfection. This could have been an effective way to assess both of question ratiometrically on a cell-by-cell basis via FACs. The authors did not do any of these.

Instead the authors designed a construct to (GFP-IC1-mCherry) that could have potentially provided a third (in some ways ideal) method to address this issue (i.e., by quantifying in transfer via membrane GFP+mCherry vs shedding GFP in a single reported). Quantitative ADAM10-dependency experiments with this construct has the advantage that the input stoichiometry of expression on transfected EC of GFP and mCherry signals are controlled to be precisely. However, the authors missed the opportunity to take advantage of there new construct and do studies described herein, even after they were suggested by 2 of the reviewers. They have, however, now included some minimal results as a final panel in the supplement that shows for 6 individual neutrophils following transmigration across, GFP-IC1-mCherry expressing EC. In all cases both GFP and mCherry signals are evident (with the latter at apparent lower intensity) coincident with anti-ICAM-1 staining. No quantitation (i.e., FACs) was provided and the role of ADAM10 remains unaddressed in this experiment.

Author Response

Response in the attached PDF file.
